# Gut Microbiota Associated with *Clostridioides difficile* Carriage in Three Clinical Groups (Inflammatory Bowel Disease, *C. difficile* Infection and Healthcare Workers) in Hospital Field

**DOI:** 10.3390/microorganisms11102527

**Published:** 2023-10-10

**Authors:** Elisa Martinez, Sebastien Crevecoeur, Carine Thirion, Jessica Grandjean, Papa Abdoulaye Fall, Marie-Pierre Hayette, Moutschen Michel, Bernard Taminiau, Edouard Louis, Georges Daube

**Affiliations:** 1Food Microbiology Lab, Fundamental and Applied Research for Animals and Health (FARAH), Department of Food Sciences, Faculty of Veterinary Medicine, University of Liege, 4000 Liège, Belgium; 2Department of Clinical Sciences, Immunopathology—Infectious Diseases and General Internal Medicine, University Hospital CHU of Liege, 4000 Liège, Belgium; 3Department of Gastroenterology, University Hospital CHU of Liege, 4000 Liège, Belgium; 4FoodChain ID, 4040 Herstal, Belgium; 5Department of Biomedical and Preclinical Sciences, Faculty of Medicine, University of Liege, 4000 Liège, Belgium

**Keywords:** *Clostridioides difficile*, bacterial microbiota, pathogenic bacteria, 16S rDNA profiling, gastrointestinal disease

## Abstract

*Clostridioides difficile* is an anaerobic spore-forming Gram-positive bacterium. *C. difficile* carriage and 16S rDNA profiling were studied in three clinical groups at three different sampling times: inflammatory bowel disease (IBD) patients, *C. difficile* infection (CDI) patients and healthcare workers (HCWs). Diversity analysis was realized in the three clinical groups, the positive and negative *C. difficile* carriage groups and the three analysis periods. Concerning the three clinical groups, β-diversity tests showed significant differences between them, especially between the HCW group and IBD group and between IBD patients and CDI patients. The Simpson index (evenness) showed a significant difference between two clinical groups (HCWs and IBD). Several genera were significantly different in the IBD patient group (*Sutterella*, *Agathobacter*) and in the CDI patient group (*Enterococcus*, *Clostridioides*). Concerning the positive and negative *C. difficile* carriage groups, β-diversity tests showed significant differences. Shannon, Simpson and InvSimpson indexes showed significant differences between the two groups. Several genera had significantly different relative prevalences in the negative group (*Agathobacter*, *Sutterella*, *Anaerostipes*, *Oscillospira*) and the positive group (*Enterococcus*, *Enterobacteriaceae*_ge and *Enterobacterales*_ge). A microbiota footprint was detected in *C. difficile*-positive carriers. More experiments are needed to test this microbiota footprint to see its impact on *C. difficile* infection.

## 1. Introduction

*Clostridioides difficile* is an anaerobic spore-forming Gram-positive bacterium. The intestinal carriage of *C. difficile* can be asymptomatic, but in cases of infection, it is associated with different clinical signs of disease that can vary from mild diarrhoea to pseudomembranous colitis. The carriage of *C. difficile* is found in 0–15% in healthy adults [1,2]. Inflammatory bowel disease (IBD) is the chronic inflammation of the gut [3]. The incidence of *C. difficile* infections (CDI) in hospitalized IBD patients is 5.5–7.6% [4,5]. In 2017, the incidence of CDI in an antibiotic-associated diarrhoea (AAD) population was 10.5–14.1%, and the incidence of CDI in a general hospital population was only 1.1–1.9% [6]. The incidence of *C. difficile* carriage in healthcare workers (HCWs) is not clear: some studies did not find carriage in any HCWs [7,8], and some studies found a percentage of 4.3% or 13% [7,9].

In the gut, 90% of the normal microbiota is composed of Firmicutes and Bacteroidetes [10,11]. The main genera of Bacteroidetes are *Prevotella* and *Bacteroides*, and those of Firmicutes are *Lactobacillus*, *Bacillus*, *Clostridium*, *Enterococcus*, *Faecalibacterium*, *Roseburia* and *Ruminococcus* [10,11]. The gut microbiota of CDI patients has a signature; studies show an increase in Proteobacteria, *Enterobacteriaceae* and *Escherichia* spp. and a decrease in the *Lachnospiraceae*, *Ruminococcaceae*, *Alistipes*, *Bacteroides* and *Prevotella* [12,13,14,15,16,17,18,19]. No studies have compared the gut microbiota of healthcare workers with another population of people who do not work at the hospital, and only one compared the gut microbiota between short-term and long-term healthcare workers [20]. The gut microbiota of IBD patients depends on whether they have active inflammation or not. The main changes in their flora are an increase in *Enterobacteriaceae*, *Prevotellaceae*, *Bacteroidaceae*, *Ruminococcus torques* and *R. gnavus* and a decrease in *Faecalibacterium* spp., *Roseburia*, *Butyricicoccus*, *Phascolarctobacterium* and *Akkermansia* spp. [21].

The first objective of this research was to evaluate the carriage of *C. difficile* in three groups of persons at risk in the hospital field. The first group was healthcare workers who were in contact with infectious patients and potentially *C. difficile*-infected patients. The second group was IBD patients who potentially had an inflammatory condition, altered microbiota, and an increased risk of CDI. The hypothesis was that this type of alteration in the microbiota predisposes patients to develop CDI. The last group was CDI patients to characterize the microbiota associated with this group and compare it with IBD patients’ microbiota. The second objective was to compare the bacterial microbiota of the three groups. The third objective was to study the global relationship between *C. difficile* carriage and the gut microbiota composition. The last objective was to assess the persistence of carriage over time in the three groups. We arbitrarily chose a monitoring time of 3 months to take in a long-term view of the evolution of carriage. The evolution of bacterial microbiota was monitored at three sampling times in CDI patients and in IBD patients.

## 2. Materials and Methods

### 2.1. Faeces Acquisition and Ethical Approval

The three groups were persons diagnosed with IBD in clinical remission (group 1, *n* = 15, 50 ± 15 years, sex ratio men/women: 0.25), persons diagnosed with CDI (group 2, *n* = 15, 63 ± 12 years, sex ratio men/women: 0.86) and healthcare workers from the Infectious Department in the CHU hospital (group 3 *n* = 3, 41 ± 15 years, sex ratio men/women: 2). The CDI patients were diagnosed with loose stools, were glutamate dehydrogenase-positive, and provided samples from which *C. difficile* was isolated in Petri dishes. Faecal samples were obtained anonymously. All information on the patients was obtained in terms of antibiotic or probiotic use in the two months prior to recruitment (see Appendix A for patients’ information). The study was conducted according to the guidelines of the Declaration of Helsinki and approved by the Ethics Committee of the Hospital—Faculty Ethics Committee of the University of Liège, Belgium (2020/230) (date of approval, 2020). Written informed consent was obtained from all participants involved in the study. Exclusion criteria were antibiotic and probiotic use for the HCW group, active inflammatory bowel disease for the IBD patient group and no symptoms of diarrhoea for the CDI patient group. Inclusion criteria were the absence of diarrhoea for HCWs, remission status for the IBD patient group and diarrhoea and one faeces sample positive for *C. difficile*.

Once the faeces were collected in a sterile container, they were transported as quickly as possible, and the samples were stored at 4 °C during transport. The samples were treated as quickly as possible, with a maximum of 6 h between collection and analysis. Thereafter, the samples were stored at −80 °C. For logistical reasons, some samples were stored at −80 °C and analysed later. The samples were classified according to the Bristol stool scale [22] (see Appendix A).

Participation at the three sampling times varied across the groups: all HCWs participated every time, but for IBD patient and CDI patient groups, the levels of participation were not the same: IBD-T1 (*n* = 14); IBD-T2 (*n* = 8); IBD-T3 (*n* = 10); CDI-T1 (*n* = 14); CDI-T2 (*n* = 12); CDI-T3 (*n* = 12). For IBD and HCW groups, samples were collected every three months. For the CDI patient group, the first two samples were collected during the disease, and the third one was collected when they were in remission 3 months later.

### 2.2. Isolation and Confirmation of C. difficile Presence

Direct and indirect enrichment proceeded for each faeces sample with the protocol described by Rodriguez et al. (2012) [23]. A *C. diff* quick check complete^®^ test (Abbott, Wavre, Belgium) for toxins and glutamate dehydrogenase antigen detection (GDH) was performed on each faeces sample (see Table 1). Confirmation with a specific qPCR test using 16S rDNA on DNA extracted from faeces was performed with the protocol described by Mutters et al. (2009) [24]. When a strain was isolated, the entire 16S rDNA was sequenced.

### 2.3. DNA Extraction and 16s rDNA Amplicon Sequencing

Total DNA extractions from faeces were performed using the QIAamp PowerFecal Pro DNA kit (Qiagen, Antwerp, Belgium) according to the type of sample and the manufacturer’s recommendations.

With total bacterial DNA extracted from 83 samples, 16S rDNA profiling targeting the V1-V3 hypervariable region was performed as described previously [25,26]. All libraries were run with Illumina MiSeq Technology (Illumina, SY—410-1003). The protocol used was the same as that described in the work of Ngo et al. (2018) [26]. Sequence reads were processed using the MOTHUR software package V1.41.1 and Vsearch for chimera detection [27,28]. After cleaning and searching chimeras, 13,314,296 sequence reads were obtained, subsampled at 10,000 reads per sample and clustered into 129,317 OTUs (clustering threshold of 0.03). Reference alignment and taxonomic assignments were based on the SILVA database (v1.38.1) [29] of the full-length 16S rDNA sequences. The file was processed to classify the data into 536 phylotypes at the genus level. Sequencing libraries are available in the GenBank repository under the PRJNA924547 bioproject.

### 2.4. Confirmation C. difficile Isolates and Toxin Gene Detection

From *C. difficile* cultures isolated in microbiology tests, DNA extraction was carried out using the “Blood and tissue” kit (Qiagen, Antwerp, Belgium) according to the manufacturer’s recommendations. On the extracted DNA, PCR targeting *tcdA* and *tcdB* genes was performed using primers reported by Kouhsari et al., 2019 [30]. Then, PCR targeting 16S rDNA was performed according to the protocol explained by Abdelkader et al. (2021) [31]. The length of the fragment is 1500 bp, and the universal primers are *16Sr* (TAC-GGT-TAC-CTT-GTT-ACG-AC) and *16Sf* (GAG-TTT-GAT-CMT-GGC-TCA-G). Purification of the PCR product (Wizard^®^ SV Gel and PCR Clean-Up System, Promega, Leiden, Netherlands) and Sanger sequencing (GIGA, Liege, Belgium) were conducted to confirm the *C. difficile* strain.

### 2.5. Data Analysis

Three different analyses were performed: group effect (IBD patients *n* = 32; CDI patients *n* = 38; HCW *n* = 9), *C. difficile* carriage effect (*C. difficile*-positive *n* = 28; *C. difficile*-negative *n* = 51) and sampling time effect (IBD-T1 *n* = 14; IBD-T2 *n* = 8; IBD-T3 *n* = 10; CDI-T1 *n* = 14; CDI-T2 *n* = 12; CDI-T3 *n* = 12). In this analysis, the patients were integrated as a variable to avoid any bias.

First, a study of β-diversity was realized in R studio (v4.2.2) using adonis2 (global effect) and pairwiseadonis2. In this analysis, the patients were integrated as a variable to avoid any bias. Beta-dispersion was used to visualize the differences between groups using betadisper with the vegan package in R studio v4.2.2. Then, a phylogenetic study was performed on the bacterial genera OTU using Mothur (“tree.shared”) in all samples to evaluate the stability of the microbiota over time.

Second, a study of α-diversity using the InvSimpson index (diversity), Shannon index (diversity), Chao1 index (richness) and Simpsoneven index (evenness) was conducted using R version 4.2.2 with the microbiome package. One-way ANOVAs were performed on each index and each subgroup using R.

Third, linear modelling for differential abundance (linDA) was performed to evaluate the microbiome composition using the groups, times and patients as variables in R studio (V4.2.2) [32].

## 3. Results

### 3.1. Isolation and Confirmation of C. difficile Presence

In the HCW group, no samples were *C. difficile*-positive in the analysis at any of the three time points (see Table 1).

In the IBD group, only two patients were *C. difficile*-positive in the analysis (IBD 10 T2 and IBD 07 T3), and only one was positive for toxins (IBD 10 T2). This patient was asymptomatic and reported no diarrhoea (see Appendix A). Culture and genetic approaches showed the same results.

In the CDI patient group, at sampling time 1, all samples were positive except for CDI 01, which was negative and was excluded from the study. At sampling time 2, seven patients were still positive for *C. difficile* with the rapid test and qPCR. At sampling time 3, four patients were still positive for *C. difficile* with the rapid test and qPCR (see Appendix A).

The positivity of the samples was always confirmed by qPCR. Of 27 positive samples, only 16 were isolated in culture.

### 3.2. Confirmation of C. difficile Strains and Toxin Research

Table 2 reports the results of *tcdA* and *tcdB* PCR and Sanger sequencing. In the IBD patient group, two isolates of *C. difficile* carried the *tcdA* and *tcdB* genes. In the CDI patient group, seventeen isolates of *C. difficile* were obtained, and eleven out of the seventeen carried the *tcdA* and *tcdB* genes.

### 3.3. Group Effect on Bacterial Microbiota

Five samples were excluded from the statistical analysis. Samples CDI 01 T1 and T2 were excluded because CDI 01 was not positive at sampling time 1. Patient IBD 01 T1-T2-T3 was removed from the analysis because the amplicon profiling revealed an outlier faeces profile dominated by *Lactobacillus* spp.

In the three groups, the most abundant phyla were Firmicutes (88.67% ± 8.4 HCW, 81.65% ± 14.3 IBD, 75.96% ± 25.6 CDI), Bacteroidetes (9.04% ± 8.2 HCW, 13.40% ± 11.8 IBD, 10.72% ± 14.4 CDI), Proteobacteria (0.79% ± 0.7 HCW, 2.37% ± 5.8 IBD, 6.81% ± 16.1 CDI), Actinobacteria (0.82% ± 0.8 HCW, 0.44% ± 0.7 IBD, 0.76 ± 1.2 CDI), Verrucomicrobia (0.51% ± 0.8 HCW, 1.08% ± 0.03 IBD, 1.82% ± 6.8 CDI) and Fusobacteria (0.02% ± 0.0 HCW, 0.38% ± 1.3 IBD, 3.64% ± 14.9 CDI).

In the three groups, the most abundant families were *Lachnospiraceae* (48.8% ± 15.3 HCW, 50.2% ± 23.5 IBD, 39.9% ± 27.3 CDI), *Ruminococcaceae* (22.8% ± 12.4 HCW, 13.1% ± 14.0 IBD, 7.9% ± 10.7 CDI), *Bacteroidaceae* (5.6% ± 4.4 HCW, 9.3% ± 11.0 IBD, 9.0% ± 13.6 CDI), *Clostridia*_fa (2.5% ± 1.5 HCW, 1.3%± 1.3 IBD, 0.6% ± 0.9 CDI), *Prevotellaceae* (0.4% ± 0.8 HCW, 2.2% ± 6.1 IBD, 0.6% ± 1.5 CDI), *Peptostreptococcaceae* (3.1% ± 3.0 HCW, 3.1% ± 6.6 IBD, 3.7% ± 6.0 CDI), *Streptococcaceae* (1.6% ± 4.1 HCW, 3.6% ± 8.7 IBD, 5.8% ± 9.4 CDI) *Enterococcaceae* (0.1% ± 0.3 HCW, 0.4% ± 1.4 IBD, 7.1% ± 22.5 CDI) and *Enterobacteriaceae* (0.2% ± 0.3 HCW, 0.8% ± 1.6 IBD, 6.1% ± 14.9 CDI).

In the three groups, the most abundant genera (see Figure 1A) were *Lachnospiraceae*_ge (17.45% ± 8.1 HCW, 20.58% ± 16.4 IBD, 14.12% ± 12.4 CDI), *Faecalibacterium* (16.63% ± 13.4 HCW, 9.85%± 10.6 IBD, 5.11% ± 8.8^b^ CDI), *Blautia* (9.76% ± 2.8 HCW, 13.17% ± 8.9 IBD, 13.31% ± 12.4 CDI), *Bacteroides* (5.12% ± 4.6 HCW, 9.51% ± 11.0 IBD, 9.00% ± 13.6 CDI), *Agathobacter* (4.50% ± 4.1 HCW, 1.97% ± 2.9 IBD, 0.32 ± 1.2 CDI), *Clostridia*_ge (2.24% ± 1.7 HCW, 1.46% ± 1.3 IBD, 0.63% ± 0.85 CDI), *Enterococcus* (0.00% ± 0.0 HCW, 0.28%± 1.4 IBD, 7.06% ± 9.3 CDI) and *Streptococcus* (0.24% ± 0.2 HCW, 2.38% ± 6.6 IBD, 5.72% ± 9.5 CDI).

In the study of α-diversity, only the InvSimpson index was significantly different between HCW and CDI (see Figure 1B). In the study of β-diversity, the global group effect was significantly different (*p* = 0.005). The HCW group vs. IBD patient group (*p* = 0.002) and IBD patient group vs. CDI patient group (*p* = 0.001) showed significant differences. In Figure 1C, a visualization of β-dispersion is shown. With linDA models, five genera were associated with the CDI group, and thirteen genera were associated with the IBD patient group. The results are shown in Figure 1D. The results of linDA between the HCW group and the CDI patient group are shown in Appendix A.

### 3.4. C. difficile Carriage Effect on Bacterial Microbiota

The more abundant genera in the *C. difficile*-negative group compared to the *C. difficile*-positive group (see Figure 2A) were *Lachnospiraceae*_ge (18.5% versus 14.4%), *Blautia* (13.7% versus 11.3%) and *Faecalibacterium* (10.3% versus 4.6%). The more abundant genera in the *C. difficile*-positive group compared to the *C. difficile*-negative group were *Bacteroides* (9.5% versus 8.4%), *Enterococcus* (9.8% versus 0.3%), *Streptococcus* (5.7%, 2.7%), *Dorea* (4.9% versus 1.7%), *Lachnoclostridium* (5.0% versus 1.8%) and *Escherichia*-*Shigella* (4.2% versus 0.4%). 

In the study of α-diversity, only the InvSimpson index and Shannon index were significantly different between the two groups (see Figure 2B). In the study of β-diversity, the global group effect was significantly different (*p* = 0.04). In Figure 2C, a visualization of β-dispersion is shown. With linDA models, four genera were associated with the *C. difficile*-positive group, and twenty-one genera were associated with the *C. difficile*-negative group. The results in Table 3 are shown in Figure 2D. In Table 3, the mean relative abundance of the main genera (>0.5%) in *C. difficile*-positive and *C. difficile*-negative groups is presented. In Appendix A, the details of the linDA analysis are shown.

### 3.5. Sampling Time Effect on Bacterial Microbiota

In the study of bacterial microbiota evolution over time (See Figure 3A), the α-diversity study showed statistical differences in InvSimpson between T2 and T3 in the CDI group (See Figure 3B1) and in Chao1 between T1 and T3 and in evenness between T1 and T2 in the IBD group (See Figure 3B2). There were no statistical differences in β-diversity or in differential abundance in the CDI group. In the study of β-diversity, there were statistical differences between T1 and T2 (*p* = 0.01172) and between T1 and T3 (*p* = 0.003) in the IBD group. The differential abundance analysis of the IBD group is shown in Appendix A.

In Figure 4, a representation of the stability of each patient is shown in a dendrogram using the “tree.shared” formula. No phylogenetic groups were marked by this representation. Some samples are clustered as a function of the group, but most of the time, the samples are grouped as a function of the patient. In Appendix A, the details of gut microbiota profiling of the nine HCW samples, IBD patient samples and CDI patient samples are shown.

## 4. Discussion

The objectives of this research were to evaluate the carriage of *C. difficile* in three groups of persons at risk in the hospital field, to evaluate the persistence of carriage over time, to study the relationship between *C. difficile* carriage and the gut microbiota composition and to compare the bacterial microbiota in the three clinical groups.

In the HCW group, no patients were *C. difficile*-positive at any sampling time. This population is in contact with *C. difficile* cases in the hospital. Unfortunately, the level of involvement was below the desired level. With only three people at three different times (*n* = 9), these results cannot be considered representative of the carriage of *C. difficile* in HCWs in the hospital field. In a hospital in Sweden, 0% of HCWs had *C. difficile*-positive stools and were in contact with *C. difficile* patients [33]. Even though it was a small number of subjects, it is the first preliminary study that examines the gut microbiota of HCWs of the Department of Infectiology of our hospital over time. In Appendix A, the details of gut microbiota profiling of the nine samples are shown.

In the IBD patient group, two patients were *C. difficile*-positive during the study. This asymptomatic carriage was transient over time. Patient IBD 07 was positive at the third time point, and patient IBD 10 was positive at the second time point. The sample of IBD 10 T1 (sampling time before positivity) had a microbiota similar to CDI patients (see dendrogram). In patient IBD 10, the abundance of *Enterococcus* (7.78% T1; 0.10% T2; 0.10% T3), *Escherichia*/*Shigella* (1.48% T1; 0.53% T2; 1.23% T3), *Erysipelatoclostridium* (42.22% T1; 0.67% T2; 0.53% T3) and *Erysipelotrichaceae* (35.52% T1; 0.04% T2; 0.95% T3) was similar to CDI patient microbiota. Before the asymptomatic carriage of *C. difficile* (T1), the gut microbiota was modified, with an increase in certain bacteria that are present in *C. difficile*-positive patients. Unfortunately, for patient IBD 07, only two samples were given: sampling time 1 (before positivity) and sampling time 3 (during positivity). In patient IBD 07, the abundance of *Faecalibacterium* (15.42% T1; 1.91% T3), *Lachnospiraceae*_ge (12.35% T1; 45.70% T3), *Streptococcus* (1.64% T1; 0.10% T3), *Blautia* (25.03% T1; 6.00% T3) and *Dorea* (2.3% T1; 4.96% T3) was concordant with CDI patient microbiota. During asymptomatic carriage (T3), some genera were reduced compared to T1 (*Blautia*, *Faecalibacterium)*, as described by Martinez et al., 2022. Before asymptomatic carriage (T1), some genera were increased compared to T3 (*Streptococcus* spp.), and some were reduced (*Lachnospiraceae*_ge and *Dorea*). These differences were observed during the *C. difficile* infection [34]. The IBD patient microbiota is dynamic, and depending on sampling, the microbiota may vary [35]. Our results are valid only for microbiota studied from faeces.

In the CDI patient group, at the three sampling times, seven patients were positive once, three patients were positive twice, and four patients were positive all three times. Several genera were statistically different between the IBD and CDI groups. Some genera were significantly decreased in the CDI patient group in comparison with the IBD patient group: GCA.900066575, *Lachnospirales*_ge, *Oscillospira*, UCG.003, *Paraprevotella*, *Clostridiales*_ge, *Colidextribacter*, *Holdemanella*, *Sutterella*, *Monoglobus*, *Agathobacter*, *Alistipes*, *Clostridia_vadinBB60_group_ge.* These modifications were already described for *Agathobacter*, *Alistipes*, *Clostridiales*_ge, *Paraprevotella* and *Sutterella* in a recent study that identified the main significantly different genera of the gut microbiota of CDI patients versus healthy people [34]. Some genera were significantly decreased in the CDI patient group in comparison with the HCW group and IBD patient group: *Sutterella* and *Agathobacter*. Some genera were significantly increased in the CDI patient group in comparison with the IBD patient group: *Clostridioides*, *Enterococcus*, *Enterobacteriaceae_ge*, *Enterobacterales*_ge and *Lactobacillales*_ge. Martinez et al. (2022) described similar modifications in the gut microbiota of CDI patients versus healthy people [34].

This work has limitations. The level of involvement in the HCW group was below the expectation, which prevents a real statistical analysis for this population. 

In the positive *C. difficile* carriage group, several genera were significantly decreased: *Lachnospiraceae*_NK4A136, *Colidextribacter*, *Monoglobus*, UCG.003, *Oscillibacter*, *Oscillospiraceae*_ge, *Oscillospira*, *Coriobacteriales*_ge, *Marvinbryantia*, *Romboutsia*, *Ruminococcaceae*_ge, *Fusicatenibacter*, *Butyricicoccus*, *Clostridia*_ge, *Sutterella*, *Lachnospiraceae*_UCG.004, *Agathobacter*, *Clostridiales*_ge, *Lachnospira*, *Anaerostipes, Rhodospirillales_fa_ge*. These modifications were described for *Ruminococcaceae*_ge, *Oscillospira*, *Fusicatenibacter* and *Anaerostipes* in a recent study that identified the main significantly different genera of the gut microbiota of CDI patients versus healthy people [34]. In the positive *C. difficile* carriage group, several genera were significantly increased: *Enterococcus*, *Enterobacteriaceae*_ge, *Enterobacterales*_ge, *Clostridioides*. These modifications were described by Martinez et al., 2022 [34].

In positive CDI samples, several genera were increased: *Veillonella* spp.; *Enterococcus* spp.; *Streptococcus* spp.; *Escherichia*-*Shigella*; *Enterobacteriaceae*_ge. *Veillonella* spp. is associated in the literature with CDI [34], irritable bowel syndrome [36] and colorectal cancer [36]. *Enterococcus* spp. are present in human vaginal secretions, human milk, fermented foods and dairy products [37,38,39,40]. In the species analysis, *Enterococcus faecalis* (29.6% of positive samples vs. 3.8% of negative samples) and *Enterococcus faecium* (22.2% of positive samples vs. 13.5% of negative samples) were the two species present in the samples. *E. faecium* and *E. faecalis* are controversial bacteria. They are commensal bacteria of the intestinal flora. Recently, it was shown that *E. faecalis* and *E. faecium* are potentially pathogenic bacteria due to their ability to adapt to new environments [34,37,40]. Additionally, resistance to vancomycin has emerged in this genus [34,37,40]. Romyasamit et al. (2020) reported that six *E. faecalis* strains have a probiotic effect and anti-*C. difficile* activity [38]. *Streptococcus* spp. are associated in the literature with CDI [34] and irritable bowel syndrome [36]. In the species analysis, *Streptococcus parasanguinis* and *Streptococcus mutant* were present in 63% and 25%, respectively, of positive CDI samples. *Streptococcus parasanguinis* is a common human commensal bacteria present in the oral cavity and plays an active role in dental plaque formation. *Streptococcus mutant* is a bacterium involved in oral infections [41,42]. In other research, *Streptococcus parasanguinis* was associated with the asymptomatic carriage of *C. difficile* [43]. *Escherichia*-*Shigella* was present in 77.8% of positive CDI samples and 46.1% of negative CDI samples. *Intestinibacter bartlettii* was present in 48.1% of both types of samples. This novel species showed decreased abundance with the use of a diabetes type 2 drug (Metformin) [44]. *Klebsiella oxytoca* (29.6% of positive samples vs. 1.9% of negative samples) and *Klebsiella pneumoniae* (25.9% of positive samples vs. 17.3% of negative samples) are pathogenic bacteria in humans. They are responsible for bronchopneumonia, urinary tract infections and septicaemia [45].

In the negative *C. difficile* carriage group, several genera were significantly increased: *Lachnospiraceae*_NK4A136, *Colidextribacter*, *Monoglobus*, UCG.003, *Oscillibacter*, *Oscillospiraceae*_ge, *Oscillospira*, *Coriobacteriales*_ge, *Marvinbryantia*, *Romboutsia*, *Ruminococcaceae*_ge, *Fusicatenibacter*, *Butyricicoccus*, *Clostridia*_ge, *Sutterella*, *Lachnospiraceae*_UCG.004, *Agathobacter*, *Clostridiales*_ge, *Lachnospira*, *Anaerostipes*, *Rhodospirillales_fa_ge*. *Lachnospiraceae* are bacterial producers of short-chain fatty acids [46]. *Blautia*, *Coprococcus*, *Dorea*, *Lachnospira*, *Oribacterium* and *Roseburia* are the main genera of the *Lachnospiraceae* family. *Eubacterium* spp., *Roseburia* spp. and *Faecalibacterium* spp. are butyrate producers and considered beneficial for human health [47]. *Eubacterium* spp. are present in the oral cavity, the intestinal tract and the environment [47]. *Eubacterium* (and also *Agathobacter rectalis* [48,49]) *rectale* was present in 57.7% of negative CDI samples and 14.8% of positive CDI samples. *Anaerostipes* spp. are also butyrate producers and lactate consumers [50]. In the species analysis, *Anaerostipes hadrus* was present in 71% of negative CDI samples and 18.5% of positive CDI samples. *Anaerostipes hadrus* is a dominant species of human colonic microbiota and is a butyrate-producing bacterium. This bacterium was associated with asymptomatic carriage of *C. difficile* in a study by Fishbein et al. [43]. One species from the *Oscillospira* genus is associated with leanness and health [51], and *Oscillospira* spp. are reduced in patients suffering from Crohn’s disease [52]. *Lachnospiraceae*, *Coprococcus* spp., *Faecalibacterium* spp. *Fusicatenibacter* spp. and *Anaerostipes* spp. are associated with CDI in the literature [34]. 

The effect of antibiotics (ATB) was studied (see Appendix A). Fourteen samples were collected during antibiotic use. In this group, a statistical difference in beta-diversity was found between the antibiotic group and the non-antibiotic group (*p* value < 0.05). Using the linDA model with antibiotics and patients as variables, only one bacterial genus was identified as associated with the ATB group (*Enterococcus*), and four bacterial genera were identified as associated with the non-ATB group (*Flavonifractor*, *Colidextribacter*, *Lachnospiraceae*_UCG.004, *Dorea*, *Lachnospiraceae*_FCS020_group) (see Appendix A). The modifications observed with antibiotic use had little impact on the bacterial microbiota. This may be due to the small sample size taken when the antibiotic was taken or to the fact that we did not consider the temporal impact on the bacterial microbiota [53]. *Enterococcus* was associated with the ATB group, as well as positive *C. difficile* carriage and the CDI patient group. *Colidextribacter* and *Lachnospiraceae*_UCG.004 were associated with negative *C. difficile* carriage and the non-ATB group. *Dorea* was associated with healthy people in a recent study [34].

## 5. Conclusions

These results confirm previous studies of the gut microbiota of CDI patients. Changes in the microbiota in IBD patients with *C. difficile* seem to occur in the same direction. This is the first work that studies carriage over time in three different populations: healthcare workers, inflammatory bowel disease patients and *C. difficile* patients. A microbiota footprint was detected in *C. difficile*-positive carriers, and some genera could be beneficial by protecting against the colonization of *C. difficile*. More experiments are needed to test this microbiota footprint to see its impact on *C. difficile* infection. If an impact is confirmed, an early screening of the gut microbiota at hospital admission could identify patients at higher risk of CDI.

## Figures and Tables

**Figure 1 microorganisms-11-02527-f001:**
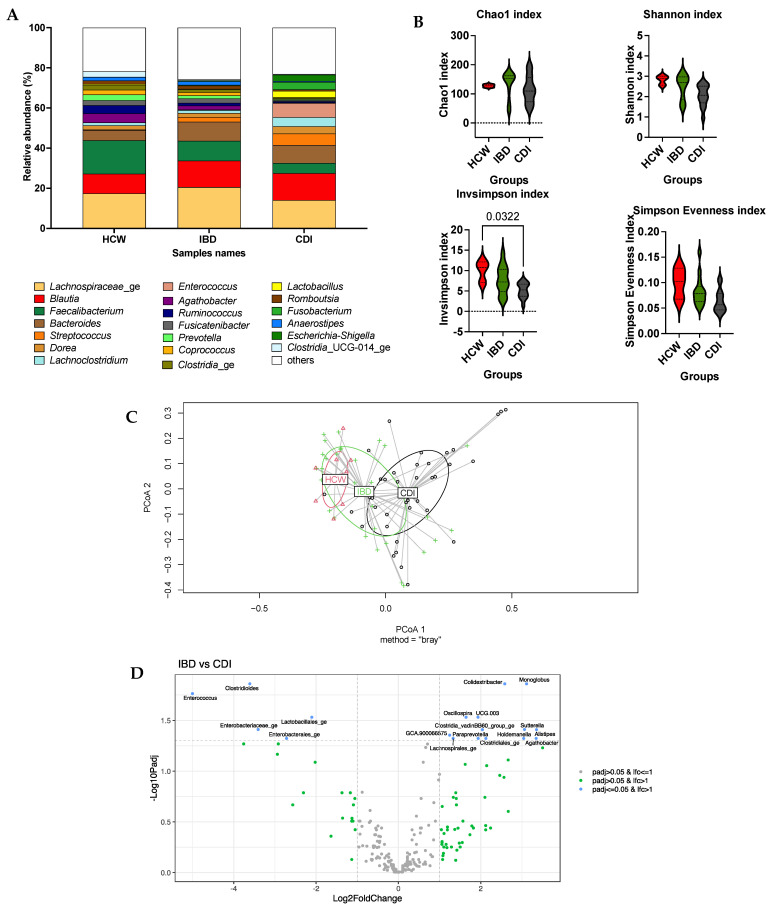
(**A**) Mean cumulative relative abundance of bacterial genera for three clinical groups: HCW group (*n* = 9), IBD patient group (*n* = 32) and CDI patient group *(n* = 38) assessed by 16S rDNA profiling using Prism v9.1.1. (**B**) Analysis of α-diversity using Chao1, Shannon, InvSimpson and Simpson indexes in three clinical groups in R studio v4.2.2. Graphs were created in Prism v9.1.1. The repetitive data from one patient are expressed as one mean value. (**C**) Visualization of β-dispersion using betadisper in Rstudio v4.2.2. Betadisper analysed beta-dispersion on the basis of the Bray–Curtis dissimilarity matrix. Dissimilarity (beta-dispersion) between groups was assessed using the Adonis2 test. The HCW group is represented in red, the IBD group is represented in green, and the CDI group is represented in black. (**D**) Linear model for differential abundance in Rstudio v4.2.2 (linDA) using groups and patients as variables. Here, data are presented as log_2_fold change in the IBD group vs. CDI group. Two thresholds are present: the first one is fixed at -log_10_(*padj* > 0.05), and the second one is fixed at −1 and +1 log_2_fold change. The grey dots are genera with *padj* > 0.05 and with an effect size ≤ 1; the green dots are genera with *padj* > 0.05 and with an effect size > 1; the blue dots are genera with *padj* ≤ 0.05 and with an effect size > 1. The bacterial genera associated with the IBD patient group are on the right side of the graph (positive axis), and the bacterial genera associated with the CDI patient group are on the left (negative axis).

**Figure 2 microorganisms-11-02527-f002:**
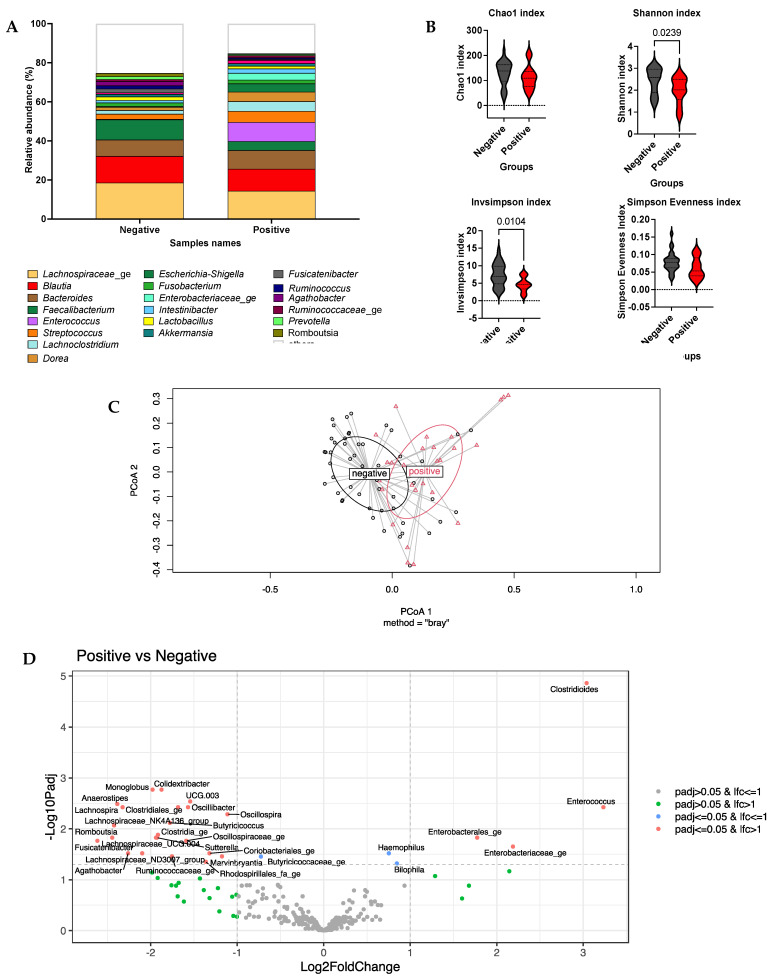
(**A**) Mean cumulative relative abundance of bacterial genera for three clinical groups: *C. difficile*-negative (*n* = 51) and *C. difficile*-positive (*n* = 28), assessed by 16S rDNA profiling using Prism v9.1.1. (**B**) Analysis of α-diversity using Chao1, Shannon, InvSimpson and Simpson indexes in two clinical groups in R studio v4.2.2. Graphs were performed in Prism v9.1.1. The repetitive data from one patient are expressed as one mean value. (**C**) Visualization of β-dispersion using betadisper in Rstudio v4.2.2. Betadisper analysed beta-dispersion on the basis of the Bray–Curtis dissimilarity matrix. Dissimilarity (beta-dispersion) between groups was assessed using the Adonis2 test. The *C. difficile*-positive group is represented in red; the *C. difficile*-negative group is represented in black. (**D**) Linear model for differential abundance in Rstudio v4.2.2 (linDA) using *C. difficile* carriage and patients as variables. Here, data are presented as log_2_fold change in the *C. difficile*-positive group vs. *C. difficile*-negative group. Two thresholds are present: the first one is fixed at -log_10_(padj > 0.05), and the second one is fixed at −1 and +1 log_2_fold change. The grey dots are genera with *padj* > 0.05 and with an effect size ≤ 1; the green dots are genera with *padj* > 0.05 and with an effect size > 1; the blue dots are genera with *padj* ≤ 0.05 and with an effect size ≤ 1 and the red dot are genera with *padj* ≤ 0.05 and with an effect size > 1. The bacterial genera associated with positive *C. difficile* carriage are on the right side of the graph (positive axis), and the bacterial genera associated with negative *C. difficile* carriage are on the left (negative axis).

**Figure 3 microorganisms-11-02527-f003:**
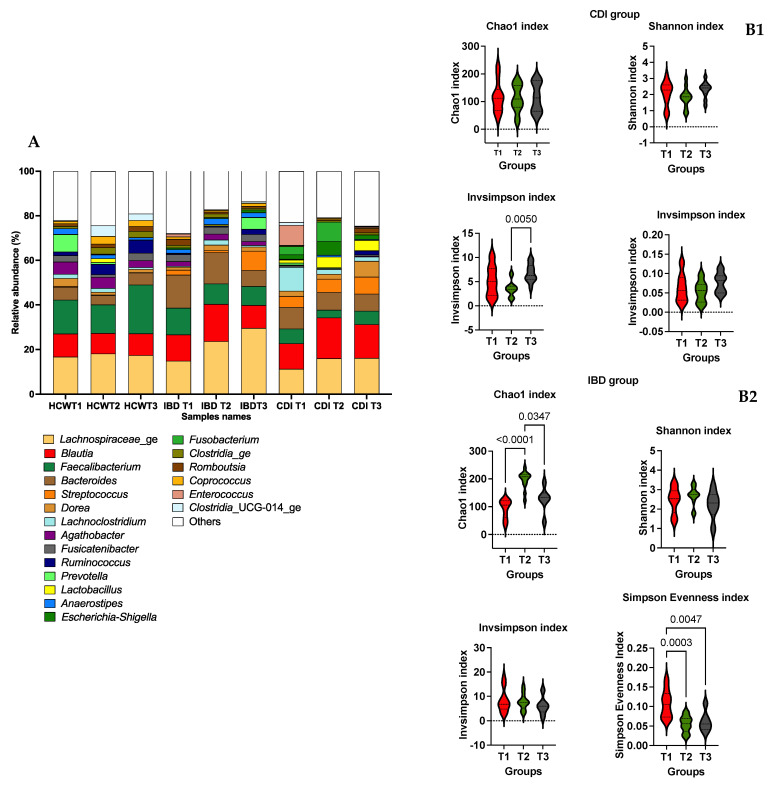
(**A**) Mean cumulative relative abundance of bacterial genera for three clinical groups: IBD patient group (IBD-T1, *n* = 14; IBD-T2, *n* = 8; IBD-T3, *n* = 10) and CDI patient group (CDI-T1, *n* = 14; CDI-T2, *n* = 12; CDI-T3, *n* = 12), assessed by 16S rDNA profiling using Prism v9.1.1. (**B1**) Analysis of α-diversity using Chao1, Shannon, InvSimpson and Simpson indexes over time in CDI patient group in R studio v4.2.2. Graphs were created in Prism v9.1.1. (**B2**) Analysis of α-diversity using Chao1, Shannon, InvSimpson and Simpson indexes over time in IBD patient group in R studio v4.2.2. Graphs were created in Prism v9.1.1.

**Figure 4 microorganisms-11-02527-f004:**
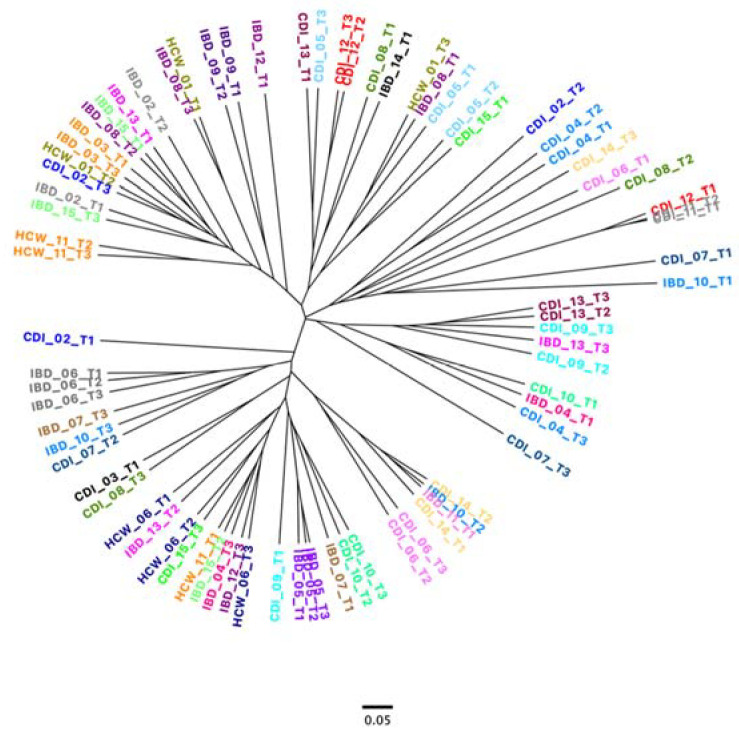
Dendrogram for three clinical groups at the three sampling times for bacterial genera using Mothur (version 1.48.0) using “tree.shared” formula. Samples from the same patient are stained with the same colour.

**Table 1 microorganisms-11-02527-t001:** Results of microbiological and genetic analyses of stool samples.

	GDH ^a^	Toxins ^a^	qPCR *C. diff*	Direct ^b^	Indirect ^b^
IBD patients					
T1	0/15	0/15	0/15	0/15	0/15
T2	1/9	1/9	1/9	1/9	1/9
T3	1/11	0/11	1/11	0/11	1/11
CDI patients					
T1	14/15	8/15	14/15	9/15	9/15
T2	7/13	2/13	7/13	4/13	3/13
T3	4/12	1/12	4/12	1/12	1/12
HCWs					
T1	0/3	0/3	0/3	0/3	0/3
T2	0/3	0/3	0/3	0/3	0/3
T3	0/3	0/3	0/3	0/3	0/3

^a^ *C. diff* quick check complete (Techlab). ^b^ Microbiological cultures.

**Table 2 microorganisms-11-02527-t002:** Main characteristics of *C. difficile* strains isolated from faeces.

Samples	Sanger Sequencing	*tcdB*	*tcdA*
IBD 10 T2	*C. difficile*	Y	Y
IBD 07 T3	*C. difficile*	Y	Y
CDI 02 T1	*C. difficile*	N	N
CDI 02 T2	*C. difficile*	N	N
CDI 03 T1	*C. difficile*	N	N
CDI 04 T1	*C. difficile*	Y	Y
CDI 05 T3	*C. difficile*	N	N
CDI 06 T1	*C. difficile*	Y	Y
CDI 07 T1	*C. difficile*	Y	Y
CDI 07 T2	*C. difficile*	Y	Y
CDI 07 T3	*C. difficile*	Y	Y
CDI 08 T1	*C. difficile*	Y	Y
CDI 08 T2	*C. difficile*	Y	Y
CDI 09 T1	*C. difficile*	N	N
CDI 10 T1	*C. difficile*	N	N
CDI 11 T2	*C. difficile*	Y	Y
CDI 15 T1	*C. difficile*	Y	Y

Legend: Y: Yes; N: No.

**Table 3 microorganisms-11-02527-t003:** Mean relative abundance of main genera (>0.5%) in *C. difficile*-positive and *C. difficile*-negative groups.

Genera	*C. difficile*-Negative	*C. difficile*-Positive	*P adj* (LinDA Models)
*Lachnospiraceae*_ge	18.5% ± 13.2	14.4% ± 15.4%	NS (0.1478)
*Blautia*	13.7% ± 8.8%	11.3% ± 12.8%	NS (0.0922)
*Bacteroides*	8.4% ± 9.7%	9.5% ± 15.3%	NS (0.55)
*Faecalibacterium*	10.3% ± 11.1%	4.6% ± 8.9%	NS (0.2120)
*Enterococcus*	0.3% ± 1.1%	9.8% ± 26.4%	(0.0037) **
*Streptococcus*	2.7% ± 7.1%	5.7% ± 9.2%	NS (0.6089)
*Lachnoclostridium*	1.8% ± 2.9%	5.0% ± 15.7%	NS (0.9720)
*Dorea*	1.7% ± 1.9%	4.9% ± 12.7%	NS (0.1287)
*Escherichia*-*Shigella*	0.4% ± 1.0%	4.2% ± 10.5%	NS (0.0681)
*Fusobacterium*	1.9% ± 11.8%	2.0% ± 7.4%	NS (0.1309)
*Enterobacteriaceae*_ge	0.1% ± 0.2%	3.5% ± 13.7%	(0.0223) *
*Intestinibacter*	1.1% ± 2.8%	2.2% ± 5.1%	NS (0.4192)
*Lactobacillus*	1.9% ± 8.0%	1.4% ± 3.2%	NS (0.2338)
*Akkermansia*	1.3% ± 4.8%	1.3% ± 5.9%	NS (0.2690)
*Clostridium*_*sensu*_*stricto*_1	0.8% ± 1.3%	1.7% ± 3.8%	NS (0.5143)
*Fusicatenibacter*	1.9% ± 3.3%	0.5% ± 1.3%	(0.0171) *
*Ruminococcus*	1.6% ± 2.6%	0.7% ± 1.6%	NS (0.0533)
*Agathobacter*	2.1% ± 3.2%	0.2% ± 0.6%	(0.0302) *
*Ruminococcaceae*_ge	1.1% ± 1.2%	1.1% ± 4.8%	(0.0348) *
*Prevotella*	1.5% ± 5.5%	0.6% ± 1.4%	NS (0.9842)
*Romboutsia*	1.6% ± 3.4%	0.3% ± 0.9%	NS (0.0148)
*Clostridia*_ge	1.5% ± 1.4%	0.4% ± 0.5%	(0.0131) *
*Anaerostipes*	1.6% ± 1.6%	0.3% ± 0.5%	(0.0032) **
*Oscillospirales*_ge	1.2% ± 1.8%	0.6% ± 1.3%	NS (0.4386)
*Coprococcus*	1.5% ± 1.8%	0.4% ± 0.8%	NS (0.1309)
*Erysipelatoclostridium*	1.1% ± 5.8%	0.7% ± 1.2%	NS (0.9156)
*Erysipelotrichaceae*_ge	0.8% ± 4.9%	0.8% ± 2.3%	NS (0.6146)
*Subdoligranulum*	1.0% ± 1.7%	0.4% ± 1.0%	NS (0.0719)
*Clostridia*_UCG-014_ge	0.8% ± 2.3%	0.5% ± 2.6%	NS (0.9156)
*Lachnospiraceae*_NK4A136_group	1.2% ± 5.4%	0.1% ± 0.5%	(0.0085) **
*Veillonella*	0.4% ± 0.9%	0.9% ± 2.3%	NS (0.8942)

Legend: *: *p* < 0.05; **: *p* < 0.01; NS: Not significant.

## Data Availability

The data presented in this study are openly available in the Genbank repository under the PRJNA924547 Bioproject.

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
