# Peer review of "Gut Microbiota Associated with Clostridioides difficile Carriage in Three Clinical Groups (Inflammatory Bowel Disease, C. difficile Infection and Healthcare Workers) in Hospital Field"

_microorganisms, 2023, doi:10.3390/microorganisms11102527_

Round 1

Reviewer 1 Report (New Reviewer)

The manuscript entitled “Gut microbiota associated with Clostridioides difficile carriage in three clinical groups (IBD, C. difficile infection and healthcare workers) in hospital field” evaluated the C. difficile carriage and 16S rDNA profiling in three clinical groups at three different sampling times: inflammatory bowel disease (IBD), C. difficile infection (CDI) and health-care workers (HCW). Diversity analysis was realized on the three clinical groups, the positive and negative C. difficile carriage and on the three analysis periods. The results indicated that several genera were significantly different in IBD people group (Sutterella, Agathobacter) and in CDI peo-ple group (Enterococcus, Clostridioides).. The manuscript is interesting, however, it needs to be revised prior to publication.

-Key words

Please replace keywords that are in the title, basic rule.

- The manuscript needs another revision to check for spacing (spaces after words and before the citations are often missing).

Author Response

Point 1: The manuscript entitled “Gut microbiota associated with Clostridioides difficile carriage in three clinical groups (IBD, C. difficile infection and healthcare workers) in hospital field” evaluated the C. difficile carriage and 16S rDNA profiling in three clinical groups at three different sampling times: inflammatory bowel disease (IBD), C. difficile infection (CDI) and health-care workers (HCW). Diversity analysis was realized on the three clinical groups, the positive and negative C. difficile carriage and on the three analysis periods. The results indicated that several genera were significantly different in IBD people group (Sutterella, Agathobacter) and in CDI peo-ple group (Enterococcus, Clostridioides).. The manuscript is interesting, however, it needs to be revised prior to publication.

-Keywords: Please replace keywords that are in the title, basic rule.

Response 1: I modified the keywords: Clostridioides difficile, pathogenic bacteria, 16S rDNA profiling, bacterial microbiota, gastrointestinal disease.

Point 2:  The manuscript needs another revision to check for spacing (spaces after words and before the citations are often missing).

Response 2: I couldn’t find any in the text. Could you please give me an example? I checked all the citations, and the spaces were there.

Reviewer 2 Report (New Reviewer)

1.  These authors have characterized the bacteria present in stool samples from 3 groups of subjects.  These groups included healthcare workers, patients with inflammatory bowel disease, and patients with C. difficile infection.  They tried to collect stool specimens over 3 time periods.  In the patients with IBD and the healthcare workers, the specimens were separated by 3-month intervals.  In the patients with C. difficile colitis, 2 specimens were taken during the active phase of infection and 1 specimen was taken 3 months later after remission.  Bacteria was identified by 16 S rDNA profiling.  The analysis included alpha diversity and B- diversity calculations.
2.  The authors provide a great deal of information about the bacterial isolates.  Information included the phyla designation, family designation, and genera.  The analysis indicated that there were significant differences between the 3 groups when analyzing the beta diversity.  There were significant differences between the 2 clinical groups when analyzing the alpha diversity.
3.  The authors should double check the numbers listed in line 172 and in table 2.  As I read table 2, all isolates had both tcdA and tcdB genes.  The authors should check the information in line 204 and compare it to figure 1B.
4.  The results section provides an overwhelming amount of information.  Some of this information will not be understood by readers who do not work in this field.  The authors might provide an additional sentence or 2 to help the reader understand figures 1C, 1D, 2C, 2D.

5.  The results section does not consider the possible effect of antibiotics in patients with C. difficile infection.  They should make a comment about this.
6.  They also should provide a brief summary paragraph that provides a broad overview of these results.  Do some bacteria seem to protect against the development of C. difficile infection?  Do some bacteria contribute to the severity of C. difficile infections? 

7. The idea that an early screening of gut microbiota at hospital admission could identify the patient is at high risk for C. difficile infection seems very optimistic.

In several places I am not certain that the information in the text is consistent with the results table or figures.  I have noted this in the comments to the authors.

Author Response

Response to Reviewer 2: Comments

Point 1: These authors have characterized the bacteria present in stool samples from 3 groups of subjects. These groups included healthcare workers, patients with inflammatory bowel disease, and patients with C. difficile infection. They tried to collect stool specimens over 3 time periods. In the patients with IBD and the healthcare workers, the specimens were separated by 3-month intervals. In the patients with C. difficile colitis, 2 specimens were taken during the active phase of infection and 1 specimen was taken 3 months later after remission. Bacteria was identified by 16 S rDNA profiling. The analysis included alpha diversity and B- diversity calculations.

The authors provide a great deal of information about the bacterial isolates. Information included the phyla designation, family designation, and genera. The analysis indicated that there were significant differences between the 3 groups when analyzing the beta diversity. There were significant differences between the 2 clinical groups when analyzing the alpha diversity.

Response 1: Thank you

Point 2:  The authors should double check the numbers listed in line 172 and in table 2. As I read table 2, all isolates had both tcdA and tcdB genes. The authors should check the information in line 204 and compare it to figure 1B.

Response 2: I checked the table 2 and added one line (CDI 07 T3). I modified line 172: In CDI people group, seventeen isolates of C. difficile were obtained and eleven out seventeen carry the tcdA and tcdB genes.”

I verified the Figure 1.B: I modified the line 204: “In the study of a-diversity, only Invsimpson index was significant different between HCW vs CDI.

Point 3: The results section provides an overwhelming amount of information. Some of this information will not be understood by readers who do not work in this field. The authors might provide an additional sentence or 2 to help the reader understand figures 1C, 1D, 2C, 2D.

Response 3:

I added in C.1 et C.2: Betadisper analyzed beta-dispersion on the basis of Bray-Curtis dissimilarity matrix. Dissimilarity (beta dispersion) between groups was assessed using the Adonis2 test.”

I added in D.1: “The bacterial genera associated with IBD people group are on the right of the graph (positive axis) and the bacterial genera associated with CDI people group are on the left (negative axis).”

I added in D.2: “The bacterial genera associated with a positive C. difficile carriage are on the right of the graph (positive axis) and the bacterial genera associated with a negative C. difficile carriage are on the left (negative axis).

Point 4:  The results section does not consider the possible effect of antibiotics in patients with C. difficile infection. They should make a comment about this.

They also should provide a brief summary paragraph that provides a broad overview of these results. Do some bacteria seem to protect against the development of C. difficile infection? Do some bacteria contribute to the severity of C. difficile infections?  

 Response 4:

I added a paragraph in the discussion in lines 394-407: “The effect of antibiotics (ATB) has been studied. Fourteen samples were collected during antibiotics use. In this group, statistical difference in beta-diversity was found between the antibiotic group and the non-antibiotic group (P value <0.05). Using the linDA model with antibiotics and patients as variable, only one bacterial genus was identified as associated to the ATB group (Enterococcus), four bacterial genera were identified as associated to non-ATB group (Flavonifractor, Colidextribacter, Lachnospiraceae_UCG.004, Dorea, Lachnospiracea_FCS020_group).”Modifications observed with antibiotics used had little impact on the bacterial microbiota. This may be due to the low sample size taken when the antibiotic was taken, or to the fact that we did not consider the temporal impact on the bacterial microbiota [53]. Enterococcus was associated with ATB group as well as in positive C. difficile carriage and in CDI people group. Colidextribacter, Lachnospiraceae_UCG.004 were associated with negative C. difficile carriage and no-ATB group. Dorea was associated with healthy people in a recent study [34].”

Point 5: The idea that an early screening of gut microbiota at hospital admission could identify the patient is at high risk for C. difficile infection seems very optimistic.

Response 5: Yes

This manuscript is a resubmission of an earlier submission. The following is a list of the peer review reports and author responses from that submission.

Round 1

Reviewer 1 Report

The authors have addressed all my concerns.

Reviewer 2 Report

Revisions from the authors were noted and greatly appreciated. The manuscript was improved with the additional technical details and gut microbiome analysis results. However, the revisions and rebuttal were unable to eliminate/justify the major pitfalls of the current study that were primarily associated with the study design. Most importantly the authors failed to justify the small sample size (N=3 in the healthcare worker group was particularly problematic). I could not agree that merely showing statistically significant difference between groups provided any justification for the study being sufficiently powered to answer the research question(s). Other questions about the study design were also unanswered, e.g. why did the authors compare the gut microbiota of the three subject groups, particularly why were IBD patients included? Have they been shown to be at a higher risk of developing CDI; what was the justification for analyzing the gut microbiome at a three-month interval? It is also unclear to me whether all these time-course samples are used as independent samples when testing for between-group differences.

Due to the above unresolved concerns about the experimental design of the study, I am not convinced that the manuscript warrants publication.

Minor English language editing is recommended, e.g. "CDI people", "IBD people", "don't" and "three timings".